# Applications of Deep Learning to Neurodevelopment in Pediatric Imaging: Achievements and Challenges

Mengjiao Hu [1], Cosimo Nardi [2], Haihong Zhang [1] and Kai-Keng Ang [1,3,*]

1 Institute for Infocomm Research (I2R), Agency for Science, Technology and Research (A*STAR), Singapore 138632, Singapore

2 Department of Experimental and Clinical Biomedical Sciences, University of Florence—Azienda Ospedaliero-Universitaria Careggi, 50134 Florence, Italy

3 School of Computer Science and Engineering, Nanyang Technological University, Singapore 639798, Singapore

* Correspondence: kkang@i2r.a-star.edu.sg; Tel.: +65-64082678

**Abstract:** Deep learning has achieved remarkable progress, particularly in neuroimaging analysis. Deep learning applications have also been extended from adult to pediatric medical images, and thus, this paper aims to present a systematic review of this recent research. We first introduce the commonly used deep learning methods and architectures in neuroimaging, such as convolutional neural networks, auto-encoders, and generative adversarial networks. A non-exhaustive list of commonly used publicly available pediatric neuroimaging datasets and repositories are included, followed by a categorical review of recent works in pediatric MRI-based deep learning studies in the past five years. These works are categorized into recognizing neurodevelopmental disorders, identifying brain and tissue structures, estimating brain age/maturity, predicting neurodevelopment outcomes, and optimizing MRI brain imaging and analysis. Finally, we also discuss the recent achievements and challenges on these applications of deep learning to pediatric neuroimaging.

**Keywords:** pediatric; magnetic resonance imaging; neurodevelopment; deep learning

## 1. Introduction

Machine learning has achieved extraordinary achievements during the past decades. Conventional machine learning algorithms such as support vector machine and logistic regression have been widely applied to image analysis for pattern recognition and identification [1]. Yet applications of such approaches are limited by the reliance on feature extraction procedure and restrictions on high dimensionality of data. Feature extraction requires high expertise in domain knowledge to transform raw data into a different representation. Further dimension reduction techniques are required to fit the high-dimensional features to the machine learning algorithms [2]. Evolution of deep learning algorithms such as convolutional neural networks has advanced the development of machine learning to another triumph. The end-to-end framework of deep learning allows automatic feature learning of the complicated data patterns which migrates the subjectivity in feature extraction procedure. The deep architecture and nonlinear processing units empower the deep learning algorithm to deal with a vast amount of data [3,4]. Successful applications of conventional machine learning and deep learning to medical imaging have been widely reported [5,6]. Specifically, neuroimaging studies based on magnetic resonance imaging (MRI) have applied machine learning to the study of the brain in many aspects [7,8].

MRI has become a crucial diagnostic imaging technique for the study of the brain for its advantage of non-ionic and high-contrast resolution [9]. MRI relies on the nuclear magnetic resonance phenomenon, in which atomic nuclei will re-emit radio signals when placed in a magnetic field and stimulated by oscillating radio waves. Human body contains rich hydrogen nuclei and the nuclei align to the magnetic field generated by the MRI

scanner. Then, an oscillating radio frequency deviates the magnetic momentum of the nuclei from the field. When the oscillating radio pulse is removed, signals generated by the realignment of hydrogen nuclei can be detected by a reciever coil [10,11]. The most common MRI modality is the structural MRI (sMRI) which provides morphostructural information based on the concentration of hydrogen protons. sMRI measures the signals produced by aligned hydrogen protons in water molecules in the body and creates excellent contrast among different tissues. Functional MRI (fMRI) quantifies the blood oxygenation level-dependent (BOLD) signals based on the blood flow and blood oxygen changes around cells and reflects the brain activity information [12]. Resting-state fMRI (rs-fMRI) is measured when the subject is at rest while task fMRI monitors the brain function during an assigned task. Diffusion tensor imaging (DTI) estimates the motion of water molecules in the brain. The water molecules' diffusion speed and directions are restricted by tissue types and fiber architectures. DTI therefore provides information based on the quantitative anisotropy and orientation [13]. Deep learning methods have been widely applied to neuroimaging studies in adult for neuropsychiatric disorder recognition, brain tissues and structures segmentation, and clinical outcome prediction [8,14,15]. In comparison, relatively few deep learning studies have been conducted in pediatric MRI. Most previous reviews on pediatric MRI involved a large number of studies using conventional machine learning approaches instead of deep learning algorithms and some reviews focused on specific topics such as Autism [7,16,17]. To illustrate the most recent achievements of deep learning in pediatric MRI, this systematic review summarized the advanced deep learning approaches applied to multiple neurodevelopmental topics in MRI-based research in the past five years. Section 2 introduces the most commonly utilized deep learning algorithms as well as a list of available public datasets for neurodevelopment. Section 3 categorizes the recent studies into five main topics: recognizing neurodevelopmental disorders, identifying brain and tissue structures, estimating brain age/maturity, predicting neurodevelopment outcomes, and optimizing MRI brain imaging and analysis. The challenges and insights of applying deep learning to pediatric MRI are discussed in Section 4. We conclude in Section 5.

## 2. Methods

### 2.1. Deep Learning Model Architectures

Multi-layer perceptron (MLP) has the most basic architecture of deep neural networks, which is composed a stack of processing layers: an input layer, several hidden layers, and an output layer (Figure 1) [18]. The neurons in the processing layers allow nonlinear computation and empower the model to learn different representations of the training data at multiple levels of abstraction [3].

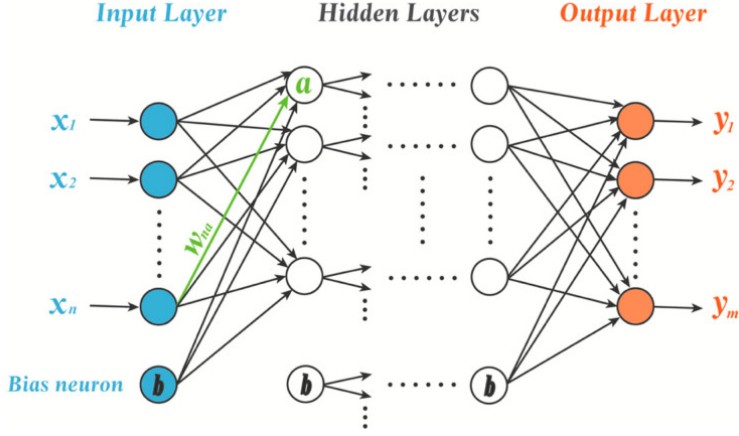

**Figure 1.** Architecture of multi-layer perceptron (MLP) [18].

Convolutional neural network (CNN) is the most widely applied deep learning algorithm for medical imaging studies. A typical CNN consists of convolutional layers with

activation functions, pooling layers, and fully connected layers (Figure 2) [2]. Convolutional layers convolve an image with different types of kernel functions to extract image features. The kernels are applied to the entire image, thus greatly reducing the number of weights to be trained compared to fully connected neural networks. Activation functions such as sigmoid and ReLu (Rectified Linear Unit) serve as nonlinear feature detectors to introduce nonlinearities to CNN. Pooling layers reduce feature map resolution with translational invariance. The combination of convolutional and pooling layers enables CNN to learn spatial hierarchies among feature patterns. Fully connected layers function as a classifier or regressor to predict the desired outcomes [2]. The weight sharing and translational invariance properties facilitate CNN the efficient and precise power on image processing tasks. Depending on the input data dimensionality, 1D, 2D, and 3D convolutional kernels can be employed. Besides the basic stacking of convolutional layers, pooling layers and fully connected layers, models with complex architectures have been developed to further improve the performance of CNN. AlexNet was the first big CNN model which showed the great potential of CNN on image recognition tasks [19]. Inception blocks utilize convolution kernels of different sizes at the same level to optimize the accuracy and computation time of the model [20]. Residual connection from a previous layer to a later layer without extra parameters solves the vanishing gradients issues and thereby make the CNN model with many layers [21]. Dense blocks formed by many convolution operations and a final pooling and connecting the input and output of each convolution are proposed to train even deeper models [22]. Many other CNN models with different architectures have been proposed. A detailed summary can be found in the review paper by Celard et al. [2].

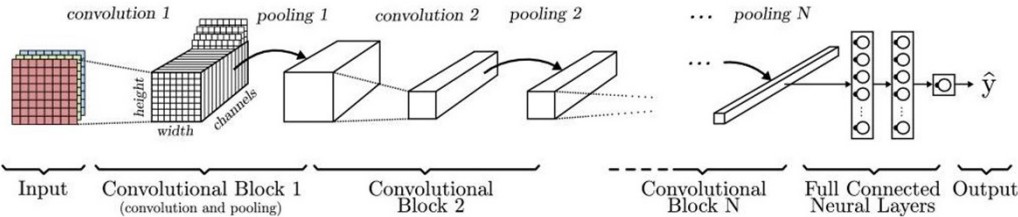

**Figure 2.** Architecture of convolutional neural networks [2].

U-net was proposed for semantic segmentation in 2015 and is still one of the most used CNN architectures for medical image segmentation. The typical U-net is composed of symmetrical encoder and decoder paths connected by skip connections (Figure 3) [23]. The model first performs a set of convolutions at the encoder side to extract features from the input data and then reconstructs the input image while including new information by transposed convolutions at the decoder side. Skip connections connect the encoder and decoder at each level. Complex architectures have also been applied to U-net to further improve its performance, for example, the Res-U-net and U-net with attention mechanism [24,25].

Auto-encoder plays a pivotal role in unsupervised deep learning. Auto-encoder follows the encoder and decoder architecture (Figure 4). The encoder aims at learning a latent representation with low dimensionality which retains only the significant information while ignoring the noise. The decoder utilizes the latent representation to reconstruct the input data. Auto-encoder provides an effective approach for feature learning in recognition tasks with unlabeled data. Variational auto-encoders are applied as generative models which randomly generate new data that are similar to the input data [2].

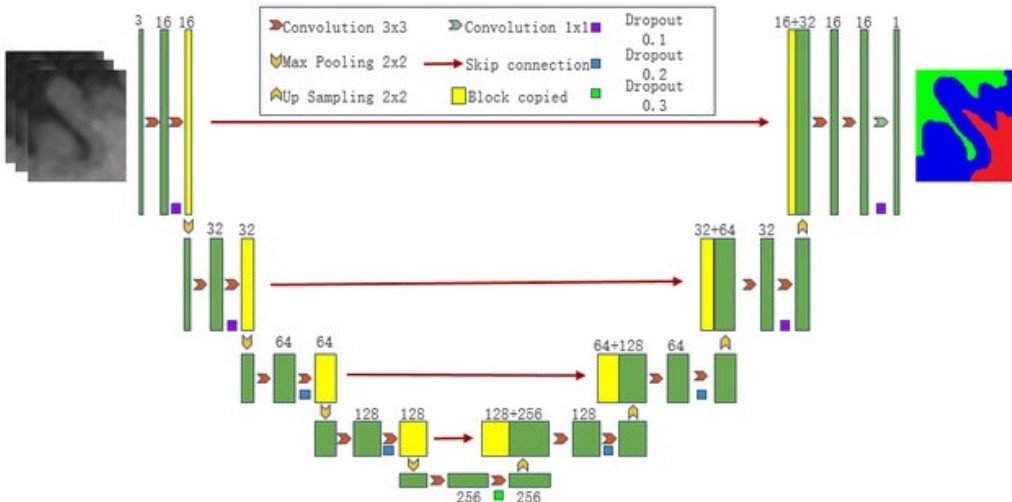

**Figure 3.** Architecture of U-net [26].

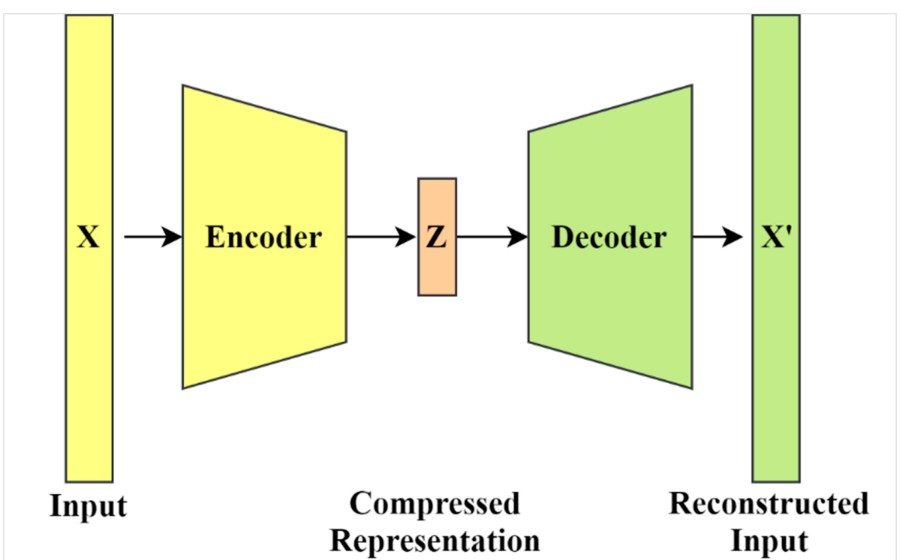

**Figure 4.** Architecture of auto-encoder [27].

Generative adversarial network (GAN) has attracted attention with its ability to model data distributions and generate realistic data since proposed in 2014 [28]. GAN consists of one generator network which captures the data distribution in real images and generates a fake image and one discriminator which classifies the generated fake images and real images (Figure 5). Two networks are trained alternatively in a competitive manner. A large number of variations of GAN have been proposed and applied to object detection, localization, segmentation, data augmentation, and image quality improvement tasks [29]. A review paper [30] introduced various architectures of GAN and their applications in medical imaging.

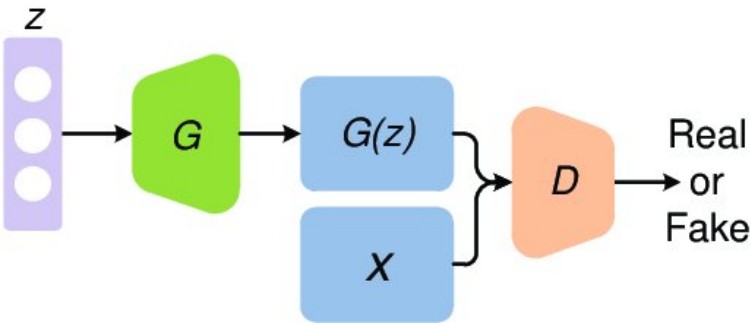

**Figure 5.** Architecture of generative adversarial networks (GAN) [29].

*2.2. Public Datasets and Repositories*

Sample size is one of the most critical issues for training a deep learning algorithm as the number of trainable parameters grows exponentially with deep architectures. However, data collection is expensive and time-consuming for medical images. Fortunately, more and more data repositories and data-sharing platforms are available recently, making it possible to conduct medical imaging studies on a large scale. Table 1 lists the available public datasets and repositories involved in the studies reviewed in this manuscript. Some repositories collect data from multiple independent sites and provide a large number of subjects. The Autism Brain Imaging Data Exchange (ABIDE) dataset and IMaging-PsychiAtry Challenge (IMPAC) dataset focus on autism spectrum disorder (ASD) recognition and provide data of subjects with ASD and healthy controls. The ADHD-200 consortium collects data for attention deficit hyperactivity disorder (ADHD) patients and healthy controls. The Healthy Brain Network (HBN) dataset and Human Connectome Project Development (dHCP) project are data collections for typically developed individuals. The UNC/UMN Baby Connectome Project (BCP) collects data of infants and pre-school age children. Other datasets including a large number of participants such as UK Biobank and International Consortium for Brain Mapping (ICBM) involve healthy controls as well as patients with various neurodevelopmental disorders at all ages.

**Table 1.** Public datasets.

| Dataset | No. of Sites/Projects | Population | Technique | Citation |
|---|---|---|---|---|
| Autism Brain Imaging Data Exchange I (ABIDE I) | 17 independent imaging sites | 539 subjects with ASD and 573 healthy controls (age 7–64 years) | sMRI, rs-fMRI | [31] |
| Autism Brain Imaging Data Exchange II (ABIDE II) | 19 independent imaging sites | 521 subjects with ASD and 593 healthy controls (age 5–64 years) | sMRI, rs-fMRI, DTI | [32] |
| IMaging-PsychiAtry Challenge (IMPAC) | - | 549 subjects with ASD 601 healthy controls (age 0–80 years) | sMRI, rs-fMRI | [33] |
| ADHD-200 Consortium | 8 independent imaging sites | 285 subjects with ADHD 491 healthy controls (age 7–21 years) | sMRI, rs-fMRI | [34] |
| UK Biobank | - | 500,000 subjects (age 40–69 years) | sMRI, rs-fMRI, DTI | [35] |

**Table 1.** *Cont.*

| Dataset | No. of Sites/Projects | Population | Technique | Citation |
|---|---|---|---|---|
| National Database for Autism Research (NDAR) | hundreds of research projects | 117,573 subjects by age (57,510 affected subjects and 59,763 control subjects) | sMRI, rs-fMRI, DTI | [36] |
| Open fMRI | 95 datasets | 3375 subjects across all datasets | sMRI, rs-fMRI, task fMRI | [37] |
| International Consortium for Brain Mapping (ICBM) | - | 853 subjects (age 18–89 years) | sMRI, rs-fMRI, DTI | [38] |
| 1000 funtional connectome | 33 independent imaging sites | 1355 subjects (age 13–80 years) | rs-fMRI | [39] |
| The Adolescent Brain Cognitive Development (ABCD) Study | - | 12,000 subjects (age 9–10 years) | sMRI, rs-fMRI, task fMRI | [40] |
| ENIGMA ADHD working group | 34 cohorts | over 4000 subjects | sMRI, rs-fMRI, DTI | [41] |
| Philadelphia Neurodevelopmental Cohort (PNC) | - | 9500 subjects (age 8–21 years) | sMRI, rs-fMRI, task fMRI, DTI | [42] |
| Healthy Brain Network (HBN) | - | 10,000 subjects (age 5–21 years) | sMRI, rs-fMRI, task fMRI, DTI | [43] |
| Human Connectome Project Development (dHCP) | - | 1350 subjects (age 5–21 years) | sMRI, rs-fMRI, task fMRI | [44] |
| The UNC/UMN Baby Connectome Project (BCP) | 2 sites | 500 subjects (age 0–5 years ) | sMRI, rs-fMRI, DTI | [45] |

Abbreviations: sMRI—structural MRI, rs-fMRI—resting-state functional MRI, DTI—Diffusion Tensor Imaging.

*2.3. Review Parameters*

The paper selection and review procedure in this study follows the preferred reporting items for systematic reviews and meta-analysis (PRISMA) guidelines [46,47]. The search terms employed were <deep learning brain MRI neurodevelopment> or <deep learning pediatric brain MRI> or <deep learning child brain MRI> or <deep learning adolescent brain MRI> to include the deep learning studies based on MRI for pediatric neurodevelopment studies. The initial search was performed on PubMed and Web of Science databases on 26 October 2022. Search engines ScienceDirect and Google Scholar were excluded due to the large number of search results returned (thousands of results).

The initial search yielded 412 papers from PubMed and 252 papers from Web of science. Following the PRISMA protocols, we performed selection and review steps in Figure 6. A total of 304 duplicate records was removed in the first step. Secondly, we examined the keywords, titles, and abstracts of the remaining 360 papers and excluded review papers, case reports, papers with foreign language (French), and animal studies. Furthermore, we identified studies with topics on adult population, genetics, maternity, and non-deep learning approaches as irrelevant and excluded them. We retrieved the full paper for 184 out of the remaining 185 studies. The full papers were further examined for eligibility and 67 studies with non-pediatric population, non-MRI modality or non-deep learning methods were removed. Then, 120 Studies were carefully reviewed and 113 of them are categorized and reported in the next chapter. The remaining 7 studies on gender prediction, functional connectivity estimation, and fascicles detection are not reported.

Three researchers independently examined the eligibility of the studies and conflict decisions were resolved by discussion. Data extracted from selected studies include but are not limited to the year of the study, clinical questions, study population, imaging techniques, preprocessing protocols and tools, deep learning approach, training and validation settings,

results, results interpretation, and limitations. Extracted information is presented and discussed in the following chapters. Specifically, risk of bias analysis was performed following the Risk Of Bias In Non-randomized Studies of Interventions [48] for (1) risk of bias due to confounding; (2) risk of bias in selection of participants into the study; (3) risk of bias in classification of interventions; (4) risk of bias due to deviations from intended interventions; (5) risk of bias due to missing data; (6) risk of bias arising from measurement of outcomes; (7) risk of bias in selection of reported results. Risk of bias analysis is presented in Appendix A (Table A1).

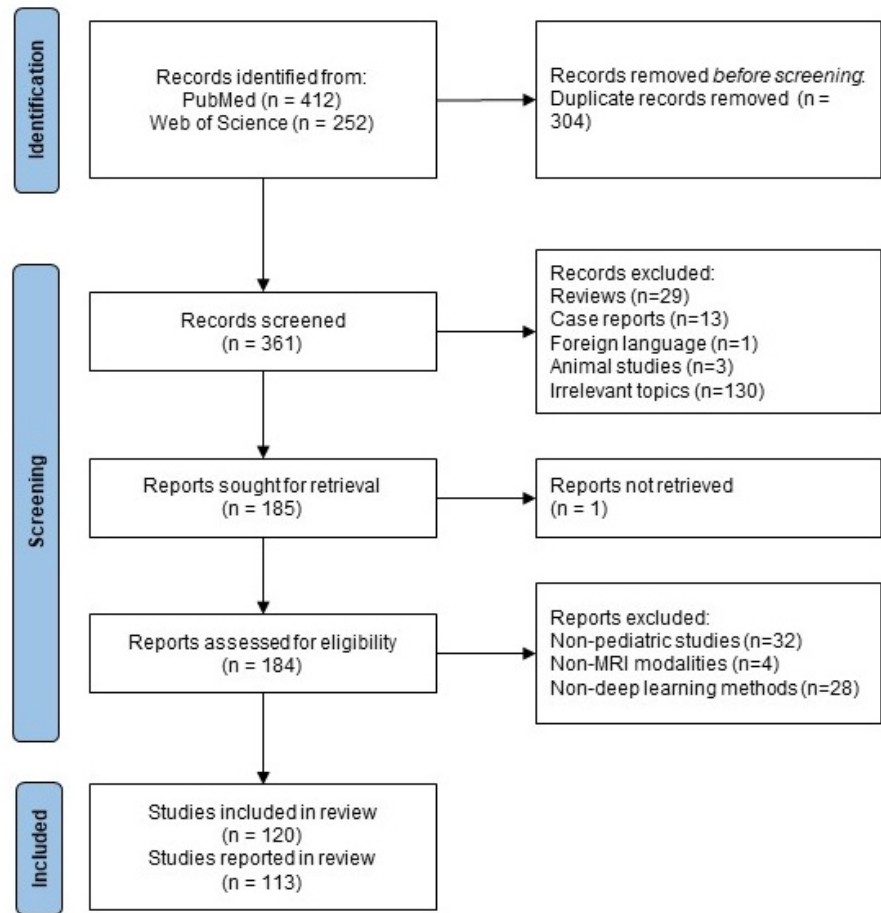

**Figure 6.** Study selection procedure.

## 3. Results

### *3.1. Recognizing Neurodevelopmental Disorders*

Neurodevelopmental disorders are common brain disorders in children, bringing a variety of challenges to the affected patients and causing great burdens to their families. Various genetic and environmental factors may perturb the developmental process and result in neurodevelopmental disorders [49].

Autism spectrum disorder (ASD) is one of the most common neurodevelopmental disorders [50]. ASD is characterized by early deficits in social interactions and communication accompanied by restricted and repetitive behaviors [49]. Review papers [7,17] summarized a selected number of studies using artificial intelligence approaches to classify ASD patients and healthy controls including both conventional machine learning methods and deep learning methods. This review listed the recent deep learning advancements using MLP, CNN, RNN, and auto-encoder models (Table 2). Rs-fMRI is widely utilized for ASD recognition. Connectomes derived from fMRI were used as inputs to MLP, CNN,

and RNN for classification [51–54]. A multimodal study [55] combined sMRI, rs-fMRI, and task fMRI.

Attention deficit hyperactivity disorder (ADHD) is another common neurodevelopmental disorder [50]. ADHD patients often suffer from hyperactivity, impulsivity, and inattention, and ADHD often continues to adulthood [56]. Previous ADHD recognition studies were summarized in the review paper [7] in conventional machine learning category and deep learning category. This review paper focuses on more recent studies utilizing deep learning approaches for ADHD detection (Table 2). Both rs-fMRI and sMRI are employed as inputs for deep learning networks.

Neurodevelopmental disorders which are less common such as cerebellar dysplasia [57], dyslexic [58], epilepsy [59,60], conduct disorder [61], disruptive behavior disorder [62], and post-traumatic stress disorder [63] are also reviewed in this study. We also include three studies for detection of posterior fossa tumors and tubers in tuberous sclerosis complex [64–66], and two studies for white matter pathway classification [67,68]. This review aims to investigate the deep learning methods utilized in various pediatric topics in an overall manner and therefore includes multiple disorders. Structural imaging techniques such as sMRI and DTI are more commonly utilized in these studies.

Overall, the selected studies are summarized in Table 2. Most studies conducted baseline comparisons using conventional machine learning approaches and reported the superior performance of deep learning approaches [53,69]. CNN dominates in the image recognition tasks. A total of 41 out of 48 neurodevelopmental disorder classification studies in this review utilized CNN approaches. Advanced CNN architectures such as inception and residual modules were employed in 2D CNN models [70–72]. Several studies trained 3D CNN with a limited number of sample size [61,69,73,74], bringing concerns on overfitting. Large-scale studies which involve thousands of training data were conducted using public datasets and repositories [55,75–78]. Multimodal studies combined features from multiple MRI modalities showed better performance than single modality [62,76].

**Table 2.** Recognizing neurodevelopmental disorders.

| Study | Year | Disorder | Population | Technique | Preprocessing | Method | Results |
|-------|------|----------|------------|-----------|---------------|--------|---------|
| [79] | 2017 | Autism | ABIDE I dataset<br>55 ASD (age 14.2 ± 3.2 years)<br>55 HC (age 12.7 ± 2.4 years) | rs-fMRI | Preprocessed Connectomes Project | MLP | Accuracy 86.36% |
| [80] | 2018 | Autism | 62 ASD 48 HC | task fMRI | FSL | MLP | Accuracy 87.1% |
| [51] | 2018 | Autism | ABIDE I dataset<br>529 ASD 571 HC | rs-fMRI | In-house pipeline | RNN | Accuracy 70.1% |
| [81] | 2018 | Autism | ABIDE I & II dataset<br>116 ASD 69 HC<br>(age 5–10 years) | sMRI, rs-fMRI | SPM8 | Deep Belief Network | Accuracy 65.56% |
| [53] | 2019 | Autism | ABIDE I & II dataset<br>210 ASD 249 HC<br>(age 5–10 years) | rs-fMRI | SPM8 | CNN | Accuracy 72.73% |
| [52] | 2019 | Autism | ABIDE II dataset<br>117 ASD 81 HC<br>(age 5–12 years) | rs-fMRI | FSL | Auto-encoder | Accuracy 96.26% |

**Table 2.** *Cont.*

| Study | Year | Disorder | Population | Technique | Preprocessing | Method | Results |
|-------|------|----------|-----------|-----------|---------------|--------|---------|
| [55] | 2020 | Autism | multi datasets: ABCD, ABIDE I, II, BioBank, NDAR, ICBM, Open fMRI, 1000 Functional Connectomes 43,838 total connectomes 1711 ASD (age 0.42–78 years) | rs-fMRI, task-fMRI | SPT, AFNI, SpeddyPP | CNN | AUROC 0.6774 |
| [82] | 2020 | Autism | YUM dataset 40 ASD (age 29.4 ± 11.6 years) 33 HC (age 30.1 ± 5.3 years) ABIDE I dataset 521 ASD (age 29.4 ± 11.6 years) 593 HC (age 30.1 ± 5.3 years) | sMRI | SPM8 | 3D CNN | Accuracy 88% (YUM) 64% (ABIDE) |
| [69] | 2021 | Autism | ABIDE I dataset 55 ASD (age 14.52 ± 6.97 years) 55 HC (age 15.81 ± 6.25 years) | rs-fMRI | Configurable Pipeline for the Analysis of Connectomes | 3D CNN | Accuracy 77.74% |
| [74] | 2021 | Autism | 50 ASD 50 HC (age 12–40 months) | task-fMRI | FSL, FEAT | 3D CNN | Accuracy 80% |
| [83] | 2021 | Autism | ABIDE I & II dataset 1060 ASD 1146 HC (age 5–64 years) | rs-fMRI | In-house pipeline | CNN | Accuracy 89.5% |
| [84] | 2021 | Autism | ABIDE I dataset 506 ASD 532 HC (age 10–28 years) | rs-fMRI | DPABI | MLP | Accuracy 78.07 ± 4.38% |
| [85] | 2021 | Autism | 52 ASD 195 HC infants (age 24 months) | MRI | iBEAT | CNN | Accuracy 92% |
| [76] | 2021 | Autism | multi datasets: ABCD, ABIDE I, II, BioBank, NDAR, Open fMRI 29,288 total connectomes 1555 ASD (age 0.42–78 years) | sMRI, rs-fMRI, task-fMRI | AFNI, SpeddyPP | CNN | AUROC 0.7354 |
| [54] | 2022 | Autism | ABIDE & UM dataset 411 HC for offline learning 48 ASD 65 HC for testing (age 13.8 ± 2 years) | rs-fMRI | Connectome Computation System | Auto-encoder | Accuracy 67.2% |
| [73] | 2022 | Autism | Preschool dataset 110 subjects ABIDE I dataset 1099 subjects | sMRI | SPM8 | CNN | AUROC 0.787 (preschool) 0.856 (ABIDE) |
| [86] | 2022 | Autism | 151 ASD 151 HC (age 1–6 years) | sMRI | In-house pipeline | 3D CNN | Accuracy 84.4% |
| [75] | 2022 | Autism | IMPAC dataset 418 ASD 497 hc (age 17 ± 9.6 years) | sMRI, rs-fMRI | In-house pipeline | MLP | AUROC 0.79 ± 0.01 |

**Table 2.** *Cont.*

| Study | Year | Disorder | Population | Technique | Preprocessing | Method | Results |
|---|---|---|---|---|---|---|---|
| [87] | 2019 | ADHD | ADHD-200 consortium 776 subjects | rs-fMRI | In-house pipeline | 3D CNN | Accuracy 69.01% |
| [88] | 2020 | ADHD | ADHD-200 consortium 262 subjects | rs-fMRI | AFNI, FSL | CNN | Accuracy 73.1% |
| [78] | 2021 | ADHD | ENIGMA-ADHD Working Group 2192 ADHD 1850 HC (age 4–63 years) | sMRI | FreeSurfer | MLP | Testing AUROC 0.60 |
| [89] | 2022 | ADHD | ADHD-200 consortium NI site25 ADHD 23 HC (age 11–22 years) NYU site: 118 ADHD 98 HC (age 7–18 years) KKI site: 22 ADHD 61 HC (age 8–13 years) PU site: 78 ADHD 116 HC (age 8–17 years) PU-1 site: 24 ADHD 62 HC (age 8–17 years) | rs-fMRI | Preprocessed Connectomes Project | Auto-encoder | Accuracy >99% |
| [90] | 2022 | ADHD | ADHD-200 consortium NI site: 28 ADHD-I 37 HC NYU site: 72 ADHD-I, 42 ADHD-C, 96 HC OHSU site: 27 ADHD-I, 13 ADHD-C, 70 HC KKI site: 16 ADHD-I, 5 ADHD-C 60 HC PU-1 site: 16 ADHD-I, 26 ADHD-C, 88 HC PU-2 site: 15 ADHD-I, 20 ADHD-C, 31 HC PU-3 site: 7 ADHD-I, 12 ADHD-C, 23 HC | rs-fMRI | DPABI | CNN | Accuracy >99% |
| [91] | 2022 | ADHD | ADHD-200 consortium Training: 69 ADHD 99HC Testing: 24 ADHD 27 HC (age 7–21 years | rs-fMRI | Athena pipeline | CNN | Testing accuracy 67% |
| [77] | 2022 | ADHD | ADHD-200 consortium 325 ADHD 547 HC (age 12 ± 3.0 years) | rs-fMRI | Athena pipeline | CNN | Accuracy 78.7 ± 4.3% |
| [92] | 2022 | ADHD | 19 ADHD (age 10.25 ± 1.94 years) 20 HC (age 10.15 ± 2.13 years) | sMRI | SPM | CNN | Accuracy 93.45 ± 1.18% |
| [93] | 2022 | ADHD | ABCD Dataset 127 ADHD 127 HC (age 9–10 years) | sMRI | ANTs | CNN | Accuracy 71.1% |
| [57] | 2018 | Cerebellar Dysplasia | 90 patients, 40 HC | sMRI | FSL, ANTs | 3D CNN | Accuracy 98.5 ± 2.41% |
| [61] | 2020 | Conduct Disorder | 60 patients (age 15.3 ± 1.0 years) 60 HC (age 15.5 ± 0.7 years) | sMRI | - | 3D CNN | Accuracy 85% |

**Table 2.** *Cont.*

| Study | Year | Disorder | Population | Technique | Preprocessing | Method | Results |
|-------|------|----------|------------|-----------|---------------|--------|---------|
| [62] | 2021 | Disruptive Behavior Disorder | ABCD Study: 550 patients, 550 HC (age 9–11 years) | sMRI, rs-fMRI, DTI | FSL | 3D CNN | Accuracy 72% |
| [58] | 2020 | Dyslexic | 36 patients, 19 HC (age 9–12 years) | task fMRI | SPM | 3D CNN | Accuracy 72.73% |
| [94] | 2020 | Embryonic Neurodevelopmental Disorders | 114 patients, 113 HC (age 16–39 weeks) | sMRI | — | CNN | Accuracy 87.7% |
| [59] | 2020 | Epilepsy | 30 patients, 13 HC | sMRI | BET | CNN | Accuracy 66–73% |
| [60] | 2020 | Epilepsy | 59 patients, 70 HC (age 7–18 years) | DTI | SPM | CNN | Accuracy 90.75% |
| [70] | 2021 | Neonatal Hyperbilirubinemia | 47patients, 32 HC (age 1–18 days) | sMRI | | CNN | Accuracy 72.15% |
| [63] | 2021 | PTSD | 33 patients (age 14.3 ± 3.3 years) 53 HC (age 15.0 ± 2.3 years) | rs-fMRI | SPM12 | MLP | Accuracy 72% |
| [64] | 2020 | Tuber | 260 patients, 260 HC | sMRI | FSL | 3D CNN | Accuracy 97.1% |
| [65] | 2022 | Tuber | 296 patients, 245 HC (age 0–8 years) | sMRI | - | 3D CNN | Accuracy 86% |
| [71] | 2020 | Tuber | 114 patients (age 5–15.3 years), 114 HC (age 6.9–15.7 years) | sMRI | In-house pipeline | CNN | Accuracy 95% |
| [95] | 2021 | Tumor | 136 patients, 22 HC (age 0–11 years) | sMRI | SPM | CNN | Accuracy 87 ± 2% |
| [72] | 2020 | Tumor | 617 patients with tumor (age 0.2–34 years) | sMRI | Pydicom | CNN | Accuracy 72% |
| [66] | 2018 | Tumor | 233 subjects | sMRI | - | Capsule Network | Accuracy 86.56% |
| [96] | 2020 | Tumor | 39 pediatric patients | sMRI | - | CNN | Accuracy 87.8% |
| [67] | 2020 | White Matter Pathways | 89 patients with focal epilepsy (age 9.95 ± 5.41 years) | DTI | FreeSurfer | CNN | Accuracy 98% |
| [68] | 2019 | White Matter Pathways | 70 HC (age 12.01 ± 4.80 years), 70 patients with focal epilepsy (age 11.60 ± 4.80 years) | DTI | FreeSurfer, FSL, NIH TORTOISE | CNN | F1 score 0.9525 ± 0.0053 |

Abbreviations: ASD—Autism spectrum disorder, HC—healthy control, ADHD—Attention deficit hyperactivity disorder, sMRI—structural MRI, rs-fMRI—resting-state functional MRI, DTI—Diffusion Tensor Imaging, MLP—Multi-layer perceptron, CNN—Convolutional neural network.

### 3.2. Identifying Brain and Tissue Structures

Identifying brain and tissue structures is of great importance in facilitating studies investigating changes in a specific region of interest. Accurate segmentation of brain tissues and structures lays the foundation for volumetric and morphologic analysis. Volumetric

analysis of gray matter, white matter, cerebrospinal fluid, and specific brain structure such as amygdala assist in computer-aided diagnosis of neurodevelopmental disorders. Localization and segmentation of brain tumor is essential for assessment of the tumor burden as well as treatment response and tumor progression [97]. Brain masking isolates the brain from surrounding tissues across non-stationary 3D brain volumes in fMRI, which is important and challenging, especially for fetal imaging [98]. Specific challenges for pediatric brain segmentation exist due to the variations in head size and shape in children compared to adults. Rapid changes in tissue contrast and low contrast to noise ratio in fetal and newborn MRIs lead to further demanding techniques [99]. This study reviews segmentation of pediatric brain tissues, structures, tumors, and masking of fetal brain (Table 3).

Most of the studies employed U-net for segmentation. Dice scores vary across studies. 3D U-net models were implemented for brain tissue and volume segmentation [25,100–102]. Transfer learning and active learning greatly reduced the number of samples that need to be labeled for training a high-quality patch-wise segmentation method [99]. FetalGAN was proposed to segment a fetal functional brain MRI using a segmentor as the generator in GAN architecture and achieved better performance than 3D U-net [98]. Adversarial domain adaptation was used to adapt a pre-trained U-net to another segmentation task in an unsupervised learning manner [103]. Transfer learning and GAN stand for the opportunity of training segmentation algorithms with weakly labeled or unlabeled data, which may greatly reduce the tedious and time-consuming process of creating groundtruth for segmentation tasks.

**Table 3.** Identifying brain and tissue structures.

| Study | Year | Structure | Population | Technique | Preprocessing | Method | Results |
|:---:|:---:|:---:|:---:|:---:|:---:|:---:|:---:|
| [104] | 2020 | Amygdala | 171 infants (age 6 months) 204 infants (age 12 months) 201 infants (age 24 months) | sMRI | - | U-net | Dice score 0.882 (6-month) 0.882 (12-month) 0.903 (24-month) |
| [105] | 2020 | Anterior Visual Pathway | 18 subjects | sMRI | - | GAN | Dice score $0.602 \pm 0.201$ |
| [106] | 2018 | Brain Mask | 10 adolescent subjects (age 10–15 years), 25 newborn subjects from dHCP dataset | sMRI | - | CNN | F1 score $95.21 \pm 0.94$ (adolescent) $90.24 \pm 1.84$ (newborns) |
| [99] | 2019 | Brain Mask | 10 adolescent subjects, 26 newborn subjects from dHCP dataset, 25 other subjects (age 0.2–2.5 years) | sMRI | - | CNN | Improve dice score after labeling a very small portion of target dataset (<0.25%) |

**Table 3.** *Cont.*

| Study | Year | Structure | Population | Technique | Preprocessing | Method | Results |
|---|---|---|---|---|---|---|---|
| [107] | 2020 | Brain Mask | 197 fetuses (gestation age 24–39 weeks) | rs-fMRI | FSL | U-net | Dice score 0.94 |
| [98] | 2020 | Brain Mask | 71 scans of fetuses | rs-fMRI | AFNI | GAN | Dice score 0.973 ± 0.013 |
| [108] | 2020 | Brain Mask | 37 healthy fetuses (gestation age 27.3 ± 4.11 weeks) 32 fetuses with spina bifida pre-surgery (gestation age 23.06 ± 1.64 weeks) 16 fetuses post-surgery (gestation age 25.69 ± 1.21 weeks) | sMRI | -N4ITK | U-net | Dice score 0.9321 (healthy), 0.9387 (pre-surgery), 0.9294 (post-surgery) |
| [101] | 2021 | Brain Mask | 214 fetuses (gestation age 22–38 weeks) | sMRI | - | 3D U-net | Testing dice score 0.944 |
| [109] | 2021 | Brain Mask | 30 subjects (ages 2.34–4.31 years) | sMRI | - | CNN | Dice score 0.90 ± 0.14 |
| [110] | 2019 | Brain Tissue | 29 subjects (age 9.96 ± 7.16 years) | sMRI | - | 3D CNN | Dice score 0.888 (gray matter), 0.863 (white matter), 0.937 (CSF) |
| [111] | 2019 | Brain Tissue | 12 fetuses (gestation age 22.9–34.6 weeks) | sMRI | - | CNN | Dice score 0.88 |
| [112] | 2019 | Brain Tissue | 95 very pre-term infants (gestation age 28.5 ± 2.5 weeks, scan at term age), 28 very pre-term infants (gestation age 26.8 ± 2.1 weeks, scan at term age) | sMRI | - | CNN | Dice score 0.895 ± 0.098 testing dice score 0.845 ± 0.079 |
| [113] | 2020 | Brain Tissue | 47 patients with pediatric hydrocephalus (age 5.8 ± 5.4 years) | sMRI | - | CNN | Dice score 0.86 |
| [114] | 2021 | Brain Tissue | 35 subjects (age 4.2 ± 0.7 years) | sMRI | - | 3D CNN | JS = 0.83 for gray matter JS = 0.92 for white matter |
| [25] | 2021 | Brain Tissue | 98 preterm infants (gestation age ≤ 32 weeks) | DTI | In-house pipeline | 3D U-net | Dice score 0.907 ± 0.041 |
| [102] | 2022 | Brain Tissue | 106 fetuses (gestation age 23–39 weeks) | sMRI | FSL | 3D U-net | Dice score 0.897 |
| [115] | 2022 | Brain Tissue | dHCP datast: 150 term (gestation age 37–44 weeks ) 50 preterm (gestation age ≤ 32 weeks, scan at term-equivalent age) | sMRI | - | CNN | Dice score 0.88 |
| [116] | 2022 | Brain Tissue | 23 infants (age 6 ± 0.5 months) | sMRI | In-house pipeline | U-net | Dice score 0.92 (gray matter), 0.901 (white matter), 0.955 (CSF) |

**Table 3.** *Cont.*

| Study | Year | Structure | Population | Technique | Preprocessing | Method | Results |
|-------|------|-----------|-----------|-----------|---------------|--------|---------|
| [117] | 2020 | Cerebral Arteries | 48 subjects (age 0.8–22 years) | sMRI | In-house pipeline | U-net | Testing dice score 0.75 |
| [118] | 2021 | Cerebral Ventricle | 200 patients with obstructive hydrocephalus (age 0–22 years) 199 HC (age 0–19 years) | sMRI | In-house pipeline | U-net | Dice score 0.901 |
| [103] | 2021 | Cortical Parcellation Network | dHCP datast: 403 infants, ePRIME dataset: 486 infants (gestation age 23–42 weeks, scanned at term-equivalent age) | sMRI | -MRITK | GAN | Dice score 0.96–0.99 |
| [119] | 2020 | Cortical Plate | 52 fetuses (gestation age 22.9–31.4 weeks) | sMRI | In-house pipeline | CNN | Testing dice score 0.907 ± 0.027 |
| [120] | 2021 | Cortical Plate | 12 fetuses (gestation age 16–39 weeks) | sMRI | -AutoNet, ITK-SNAP | CNN | Dice score 0.87 |
| [121] | 2019 | Intracranial Volume | 80 scans of fetuses (gestation age 22.9–34.6 weeks) 101 scans of infants (age 30–44 weeks) | sMRI | - | U-net | Dice score 0.976 |
| [122] | 2022 | Limbic Structure | dHCPdataset: 473 subjects (40.65 ± 2.19) | sMRI | - | CNN | Dice score 0.87 |
| [123] | 2022 | Posterior Limb of Internal Capsule | 450 preterm infants ( gestation age ≤ 32 weeks, scan at term-equivalent age) | sMRI | In-house pipeline | U-net | Dice score 0.690 |
| [124] | 2022 | Tuber | 29 subjects (age 9.96 ± 7.16 years) | sMRI | - | U-net | Testing dice score 0.59 ± 0.23 |
| [125] | 2022 | Tumor | 311 pediatric subjects | sMRI | - | U-net | Dice score 0.773 |
| [126] | 2022 | Tumor | 177 patients (age 0.27–17.87 years) | sMRI | CaPTk software | CNN | Dice score 0.910 |
| [100] | 2022 | Tumor | 122 patients (age 0.2–17.9 years) | sMRI | ANTs | 3D U-net | Dice score 0.724 |
| [97] | 2022 | Tumor | BraTS 2020 Dataset: 369 patients local dataset: 22 patients (average age 7.5–9 years) | sMRI | In-house pipeline | U-net | Dice score 0.896 |

Abbreviations: sMRI—structural MRI, rs-fMRI—resting-state functional MRI, DTI—Diffusion Tensor Imaging, CNN—Convolutional neural network, GAN—Generative adversarial network.

### 3.3. Predicting Brain Age

The brain development of children experiences a rapid and complex stage, especially for children younger than two years. Early brain development is critical for cognitive, sensory, and motor ability. Delayed brain development can lead to many neurodevelopmental disorders in children and affect their quality of life [127]. Accurate evaluation of brain development via brain age estimation based on neuroimaging is of clinical importance to understand healthy brain development and study the brain maturity deviation caused by neurodevelopmental disorders [128].

We summarized age prediction studies involved both infants and young children (Table 4). Structural MRI techniques are commonly utilized in 2D and 3D CNN models.

Study [128] using 2D CNN on DTI achieved comparison results with human experts. Study [127] demonstrated superior performance of 3D CNN compared to conventional machine learning approaches and 2D CNN. Multimodal study [129] combined sMRI, rs-fMRI, and DTI features and yielded a mean absolute error of 0.381 years for children and adolescents aged 8–21 years old. The age difference for the study population varies and thus reporting of the relative error rate is necessary for comparing different methods in different studies.

**Table 4.** Predicting brain age.

| Study | Year | Population | Technique | Preprocessing | Method | Results |
|-------|------|------------|-----------|---------------|--------|---------|
| [84] | 2017 | 115 infants (gestation age 24–32 weeks ) | DTI | In-house pipeline | CNN | MAE 2.17 weeks |
| [130] | 2019 | 317 MRI images of 112 infants age 2 weeks (8 to 35 days); 12 months (each ±2-weeks) and 3 years (each ±4-weeks). | sMRI | In-house pipeline | 3D CNN | Accuracy 98.4% classifying three age groups |
| [131] | 2019 | PNC Dataset: 857 subject (age 8–22 years) 20% as children 20% as young adult | rs-fMRII | SPM12 | MLP | Accuracy 96.64% predicting children and young adult |
| [132] | 2020 | ABIDE II dataset 382 subjects ADHD200 consortium 378 subjects | sMRI | SPM12 | 3D CNN | MAE 1.11 years (ABIDE II dataset) 1.16 years (ADHD200 consortium) |
| [127] | 2020 | 220 subjects (age 0–5 years) | sMRI | In-house pipeline | CNN | MAE 2.26 months |
| [129] | 2020 | PNC Dataset: 839 subject (age 8–21 years) | sMRI, rs-fMRI, DTI | SPM12, DPARSF, PANDA | MLP | MAE 0.381 ± 0.119 years |
| [128] | 2021 | 161 subjects (age 0–2 years) | sMRI | In-house pipeline | CNN | MAE 8.2 weeks |
| [133] | 2021 | 84 infants (age 8 days–3 years) | sMRI | In-house pipeline | CNN | Accuracy 90% |
| [134] | 2021 | 119 subjects (age 0–2 years) | sMRI | In-house pipeline | CNN | MAE 0.98 months |
| [135] | 2021 | 220 fetuses (gestation age 15.9–38.7 weeks) | sMRI | In-house pipeline | CNN | MAE 0.125 weeks |
| [136] | 2021 | 167 patients with Rolandic epilepsy (age 9.81 ± 2.55 years), 107 HC (age 9.43 ± 2.57 years) | sMRI | CAT12, SPM12 | CNN | MAE 1.05 years for HC 1.21 years for patients |
| [137] | 2022 | 524 infants (gestation age 23–42 weeks ) | sMRI, DTI | Neonatal specific segmentation pipeline | CNN | MAE 0.72 weeks (term-born) 2.21 weeks (preterm) |

Abbreviations: sMRI—structural MRI, rs-fMRI—resting-state functional MRI, DTI—Diffusion Tensor Imaging, CNN—Convolutional neural network, GAN—Generative adversarial network, MAE—mean absolute error.

### 3.4. Predicting Neurodevelopment Outcomes

The relationship between brain structure and cognitive function is complex. Research on brain activity and connectivity builds the network theory to capture the brain trajectories. It remains a challenge in the field of neuroscience to relate basic structural properties of brain to complex cognitive functions [138]. This study reviewed research on correlating brain structure and measurable neurodevelopment outcomes such as fluid intelligence, language function, and motor function (Table 5).

The ABCD dataset provides neuroimaging data including sMRI, rsfMRI, and DTI as well as cognitive assessments such as fluid intelligence and oral reading scores. Large-scale studies based on the ABCD dataset involve thousands of data and a variety of modalities to predict neurodevelopment outcomes [138–142]. CNN models were also employed to predict motor function and cognitive deficits in very preterm infants [143,144].

**Table 5.** Predicting neurodevelopment outcomes.

| Study | Year | Score | Population | Technique | Preprocessing | Method | Results |
|-------|------|-------|-----------|-----------|---------------|--------|---------|
| [143] | 2021 | Cognitive Deficits | 261 very preterm infants (gestation age ≤32 weeks , scan at 39–44 weeks postmenstrual age) | DTI, rs-fMRI | FSL | CNN | Accuracy 88.4% |
| [145] | 2020 | Fluid Intelligence | ABCD Study 8333 subjects (age 9–10 years) | sMRI | - | 3D CNN | MSE 0.75626 |
| [141] | 2021 | Fluid Intelligence | ABCD Dataset 7709 subjects (age 9–10 years) | sMRI | FSL, ANFI, FreeSuerfer | CNN | Pearson's correlation coefficient r = 0.18 |
| [138] | 2022 | Fluid Intelligence | ABCD Dataset 8070 subjects (age 9–11 years) HCP Dataset 1079 subjects (age 22–35 years) | sMRI | FreeSurfer | CNN | MSE 0.919 (ABCD Dataset) 0.834 (HCP dataset) |
| [140] | 2022 | Fluid Intelligence | ABCD Dataset 7693 subjects (age 9–11 years) | rs-fMRI | FreeSurfer | CNN | MAE 5.582 ± 0.012 |
| [142] | 2022 | Fluid Intelligence | ABCD Dataset Training: 3739 subjects, Validation 415 subjects, Testing 4515 subjects (age 9–11 years) | sMRI | FSL, ANFI, FreeSuerfer | CNN | MSE 82.56 for testing |
| [146] | 2021 | Language Scores | 31 subjects with persistent language concerns (age 4.25 ± 2.38years) | DTI | In-house pipeline | CNN | MAE 0.28 |
| [147] | 2021 | Language Scores | 37 subjects with epilepsy (age 11.8 ± 3.1years) | DTI | FSL | CNN | MAE 7.77 |

**Table 5.** *Cont.*

| Study | Year | Score | Population | Technique | Preprocessing | Method | Results |
|-------|------|-------|-----------|-----------|---------------|--------|---------|
| [144] | 2020 | Motor | 77 very pre-term infants (gestation age <31 weeks ) | DTI | ANTS | CNN | Accuracy 73% |
| [139] | 2021 | Oral Reading | ABCD Study 5252 subjects (age 9–10 years) | sMRI, DTI | - | Auto-encoder | MSE 206.5 |

Abbreviations: sMRI—structural MRI, rs-fMRI—resting-state functional MRI, DTI—Diffusion Tensor Imaging, CNN—Convolutional neural network, MAE—mean absolute error, MSE—mean squared error.

### 3.5. Optimizing MRI Brain Imaging and Analysis

Assessing imaging quality and optimizing image acquisition are significant for medical imaging analysis. Reconstruction techniques adjust the scanning parameters to maximize the image quality and control the scanning time, which is of great benefit for pediatric imaging in which many subjects cannot stay still for a long time [148]. Furthermore, some scans may be missing or with low quality due to inadequate scanning time or fail completion by the participants. Image generation algorithms synthesize pseudo-images from low-resolution image or latent space, which provide a solution to recapture missing data or rectify scans with low quality [149]. Here, we review the deep learning algorithms for image quality assessment, reconstruction, and synthesis (Table 6).

Image quality assessment tools were constructed with 2D CNN for structural MRI and DTI [150–152]. Study [153] utilized a two-stage transfer learning strategy which showed near-perfect accuracy in evaluating image quality and is capable of real-time large-scale assessment. GANs are widely applied in image generation tasks [149,154–157]. GANs showed great capability in generating synthetic images to implement missing data or improve the signal-to-noise ratio of poor quality images [24,149]. Study [148] proposed CNN models for reconstruction which reduced the scan time by 42% while maintaining image quality and lesion detectability. CNN combined with RNN also showed superior performance in improving the signal-to-noise ratio [24].

**Table 6.** Optimizing MRI brain imaging and analysis.

| Study | Year | Task | Population | Technique | Preprocessing | Method | Results |
|-------|------|------|-----------|-----------|---------------|--------|---------|
| [158] | 2020 | Image Enhancement | 131 neuro-oncology patients (age 0.4–17.1 years) | ASL | - | Auto-encoder | SNR Gain 62% |
| [159] | 2018 | Image Generation | 28 infants (scan at birth, 3 months, and 6 months) | DTI | FSL | CNN | MAE 44.4 ± 17.5 (3-month-old from neonates) 40.1 ± 10.6 (6-month-old from 3-month-old) |
| [154] | 2019 | Image Generation | 16 subjects (age 1.1–21.3 years) | sMRI | - | GAN | MAE 52.4 ± 17.6 |
| [155] | 2020 | Image Generation | 60 subjects (age 2.6–19 years) | sMRI | In-house pipeline | GAN | MAE 61.0 ± 14.1 |

**Table 6.** *Cont.*

| Study | Year | Task | Population | Technique | Preprocessing | Method | Results |
|-------|------|------|-----------|-----------|---------------|--------|---------|
| [156] | 2022 | Image Generation | ABCD Dataset: 1517 subjects (age 9–10 years) | sMRI | - | GAN | PSNR 31.371 ± 1.813 |
| [149] | 2022 | Image Generation | 127 neonates (postmenstrual age = 41.1 ± 1.5 weeks) | sMRI | ANTs | 3D GAN | RMAE 5.6 ± 1.1% |
| [157] | 2022 | Image Generation | 125 subjects (age 1–20 years) | sMRI | FSL | GAN | PSNR 28.5 ± 2.2 |
| [150] | 2019 | Image Quality Evaluation | ABIDE Dataset: 1112 subjects (age 7–64 years) | sMRI | SPM12 | CNN | Accuracy 84% |
| [153] | 2020 | Image Quality Evaluation | BCP dataset: 534 images (age 0–6 years) | sMRI | - | CNN | capable of real-time large-scale assessment with near-perfect accuracy. |
| [151] | 2021 | Image Quality Evaluation | 211 fetuses (gestation age 30.9 ± 5.5 weeks) | sMRI | In-house pipeline | CNN | Accuracy 85 ± 1% |
| [152] | 2022 | Image Quality Evaluation | ABCD Dataset: 2494 subjects (age 9–10 years) HBN Dataset: 4226 subjects (age 5–21 years) | DTI | MATRIX, FSL | CNN | Accuracy 96.61% (ABCD Dataset) 97.52% (HBN Dataset) |
| [160] | 2021 | Image Reconstruction | 20 fetuses (gestation age 23.4–38 weeks) | DTI | SVR pipeline | CNN | RMSE 0.0379 ± 0.0030 |
| [24] | 2021 | Image Reconstruction | 305 subjects (age 0–15 years) | sMRI | In-house pipeline | CNN+RNN | PSNR 27.85+/−2.12 |
| [161] | 2022 | Image Reconstruction | 107 subjects (age 0.2–18 years) | sMRI | - | CNN | image quality improved significantly by qualitative assessment |
| [148] | 2022 | Image Reconstruction | 47 subjects (age 2.3–14.7 years) | sMRI | - | CNN | Reduce scan time by 42% |

Abbreviations: sMRI—structural MRI, ASL—Arterial spin labeling, DTI—Diffusion Tensor Imaging, CNN—Convolutional neural network, GAN—Generative adversarial network, MAE—mean absolute error, PSNR—Peak signal-to-noise ratio.

## 4. Discussion

### 4.1. Advancements in Deep Learning Applied to Pediatric MRI

This study reviews pediatric MRI studies for recognition, segmentation, and prediction tasks in neurodevelopment. Throughout the review, CNN is the most commonly utilized model. Variations and advancement based on the basic architecture have been proposed to improve the performance in multi-tasks. Multi-view 2D CNN and 3D CNN have been proposed to deal with the 3D volumes in neuroimaging [57,82,84]. The multi-view 2D CNN processes 3D volumes with slices generated from sagittal, axial, and coronal sections while 3D CNN utilizes 3D kernels in the networks. Multi-branch CNN models also utilize multimodal imaging to study the brain from different perspectives. Structural

connectomes and functional connectomes were combined for age prediction in study [129] and cognitive function prediction in study [139]. Multimodal studies classified children with ASD from healthy controls using combinations of sMRI and rs-fMRI [75,76,81]. sMRI provides structural information, fMRI provides information based on brain activity, and DTI provides information regarding quantitative anisotropy and orientation. Multimodal neuroimaging allows researchers to understand the brain from different perspectives and plays an essential role in investigating the brain functional and structural changes in pediatric neurodevelopment. Variations of U-net dominate in the segmentation tasks. Dilated-Dense U-Net and U-net with attention mechanism achieved great performance in brain structure segmentation [104,120]. Meanwhile, semi-supervised learning and transfer learning initiated studies with a small number of training data [103,122]. GAN shows its superiority in image generation tasks. Variations of GANs have been proposed to synthesize pseudo-images from low-resolution images or latent space [149,155,156]. Overall, the development of computational powers has enabled deep learning models to have more complex structures and greater ability to process 3D volumes for a variety of tasks.

### 4.2. Challenges and Future Directions

### 4.2.1. Overfitting Caused by Small Sample Size

Overfitting remains a major concern for deep learning models with deep and complex architectures, especially the models with 3D structures as the number of training parameters grows exponentially with an extra dimension [2]. The sample size should also increase to train models with many parameters to avoid overfitting. Otherwise the model might be overfitted to the training data and fail to predict new data accurately. However, neuroimaging acquisition via MRI is expensive and time-consuming. Many studies are limited to a small number of training data, experiencing the risk of overfitting [162]. In our review, some studies use cross-validation to report results while some others also report results on an independent testing dataset. The testing results are important indicators of the capability to apply the trained model on unseen new data.

Data-sharing projects and platforms provide a vast amount of neuroimaging data, facilitating large-scale studies to train deep and complex models. We share a non-exhaustive list of available public datasets and repositories in Section 2. In common practice, supervised learning, in which the deep learning model is trained with labeled data is the most widely applied learning process [15,163]. Open datasets and repositories prepared data and labels in pairs where labels can be disease diagnosis, clinical outcomes, and semantic segmentation ground truth. Other than labeled data, there are tons of neuroimaging data without labels or with a limited number of labels. Unsupervised learning and semi-supervised learning show great potential in dealing with such data. Unsupervised learning utilizes training data without any labels by separating the data into different categories with automatically learned patterns during training [15,163]. Semi-supervised learning utilizes the unlabeled data to learn the feature patterns and use the labeled data to update model weights, which has yielded superior performance with a limited number of training samples in both classification and segmentation tasks [70,110]. Transfer learning accommodates another possibility for developing deep learning algorithms with a limited number of training data. Transfer learning takes advantage of models pre-trained on large datasets and fine-tunes the system with a small number of data, providing an applicable solution for neuroimaging studies with a small sample size [60,94,97].

### 4.2.2. Inconsistent Preprocessing Pipelines

Preprocessing is another challenge in pediatric neuroimaging studies. It is necessary to remove the non-brain tissue and noise in many tasks, especially for neuroimaging data of children with significant motion artifacts. However, replication and validation of results are often thus challenged by the variations in data inclusion criteria and preprocessing pipelines. The common preprocessing steps for sMRI include brain extraction, normalization to standard templates, brain tissue segmentation, and brain surface reconstruction [93].

The fMRI preprocessing steps include brain extraction, motion correction, slice time correction, distortion correction, alignment to structural images, and confounds regression [52,90]. The DTI preprocessing steps include distortion correction, Eddy current correction, brain extraction, alignment to structural images, and tensor fitting [60]. The mentioned preprocessing steps may involve multiple preprocessing softwares and adjustments may be applied to different pipelines in different studies. We listed the specified softwares and pipelines in our results. Common preprocessing softwares include SPM [164], AFNI [165], ANTs [166], FSL [167], Dpabi [168], and FreeSurfer [169]. Some studies use in-house preprocessing pipelines or did not specify the preprocessing steps. Preprocessing in single research projects may be time- and effort-consuming while variations of preprocessing pipelines restrict the replication of research results.

Standardization in data preparation and preprocessing is an urgent need for conducting large-scale neuroimaging studies. Fortunately, efforts towards standardization have been contributed by different organizations. Many data-sharing platforms employ the Brain Imaging Data Structure (BIDS) format to adopt a standardized way of organizing neuroimaging and behavioral data [170]. Furthermore, the ABIDE dataset and ADHD200 consortium release both raw and preprocessed data with shared preprocessing pipelines [31,34]. Standardization of preprocessing pipelines will greatly improve the efficacy of neuroimaging studies in the future.

### 4.2.3. Difficulty in Interpreting Deep Learning Results

Deep learning has been criticized for its "black-box nature" which poses challenges for the interpretability and explainability of trained models, and thus brings concerns to medical decision-making. The deep learning system must provide the rationale behind the decision-making process to make trustworthy predictions [171]. Various approaches have been proposed to interpret deep learning algorithms. One of the common methods is the graph-based visualization approach, which identifies the critical regions for predicting results based on activation maps derived from model weights [172,173]. Study [92] applied such an approach to identify the brain regions where children with ADHD differed from controls. The attention mechanism which focuses selectively on information of interest also plays a vital role in the interpretability of deep learning [174]. Functional connectivity differences between ADHD patients and healthy controls were identified using deep self-attention factorization in the study [90]. There are some other techniques for interpretation such as feature importance and analyzing trends and outliers in predictions. However, studies in this review have not utilized such techniques. Deep model interpretation provides crucial information for understanding brain functions and neurodevelopment, which is of great importance for pediatric neuroimaging studies. Interpretability should be one of the research focuses in future neuroimaging studies.

### 4.3. Limitations

Although some of the studies did not specify the limitations, there are some common limitations shared across individual studies. Firstly, many studies trained with a limited number of training samples, risking the bias of overfitting. The lack of independent testing results greatly restrains the generalizability of trained models to unseen data. Secondly, architectures of deep neural networks in many studies are trained in a non-exhausted exploration manner that is restricted by computational power. Thirdly, interpretation of the results is lacking in many studies and thus inhibits the interpretability and explainability of trained models. Lastly, for multi-site data which have different scanning protocols, confounding factors might cause risks of bias in the results.

This review systematically organized the most recent research on deep learning applied to pediatric MRI. However, we are unable to include the thousands of results returned by databases GoogleScholar and ScienceDirect, which remains a limitation of the study. Further investigations on unlisted studies may be applied with automatic review tools for paper selection. Keywords selected for the review are not disorder-specific and hence

may neglect some studies optimal for the inclusion criteria but not included in the initial research. Future studies on specific disorders may accommodate the limitations.

## 5. Conclusions

Deep learning plays an essential role in recent neuroimaging studies. Advancements in applications of deep learning to pediatric neuroimaging have been illustrated in this review. Complex deep learning models such as CNN and GAN have shown superior performance in neuroimaging recognition, prediction, segmentation, and generation tasks. Semi-supervised learning demonstrated great potential in the utilization of weakly labeled or unlabeled data. Challenges such as overfitting, preprocessing variations, and interpretation issues remain in many neuroimaging studies, but data-sharing platforms, standardization of preprocessing protocols, and advanced interpretation approaches have been proposed to tackle such difficulties. Future neuroimaging research on large scales will not only achieve high accuracy but also benefit the understanding of the brain functions and neurodevelopment.

**Author Contributions:** Writing—original draft preparation, M.H.; writing—review and editing, K.-K.A., H.Z. and C.N. All authors have read and agreed to the published version of the manuscript.

**Funding:** The research is supported by Institute for Infocomm Research (I2R), Agency for Science, Technology and Research (A*STAR), Singapore, and also by the A*STAR Strategic Programme Funds Project No. C211817001 Brain Body Initiative.

**Institutional Review Board Statement:** Not applicable.

**Informed Consent Statement:** Not applicable.

**Data Availability Statement:** Not applicable.

**Conflicts of Interest:** The authors declare no conflict of interest.

## Abbreviations

The following abbreviations are used in this manuscript:

| | |
|---|---|
| ABCD | The Adolescent Brain Cognitive Development |
| ABIDE | Autism Brain Imaging Data Exchange |
| ADHD | Attention deficit hyperactivity disorder |
| ASD | Autism spectrum disorder |
| ASL | Arterial spin labeling |
| CNN | Convolutional neural network |
| dHCP | Human Connectomme Project Development |
| DTI | Diffusion tensor imaging |
| fMRI | functional MRI |
| GAN | Generative adversarial network |
| HBN | Human Brain Network |
| HC | Healthy control |
| ICBM | International Consortium for Brain Mapping |
| IMPAC | Imaging Psychiatry Challenge |
| MAE | mean absolute error |
| MLP | Multi-layer perceptron |
| MRI | Magnetic resonance imaging |
| MSE | mean squared error |
| NDAR | National Dtabase for Autism Research |
| PNC | Philadelphia Neurodevelopmental Cohort |
| PRISMA | preferred reporting items for systematic reviews and meta-analysis |
| PSNR | Peak signal-to-noise ratio |
| rs-fMRI | resting-state fMRI |
| sMRI | structural MRI |

## Appendix A. Risk of Bias Analysis

Risk of bias analysis were performed following the Risk Of Bias In Non-randomized Studies of Interventions [48] for (1) risk of bias due to confounding (age, gender, scanning parameters); (2) risk of bias in selection of participants into the study (population, sample size); (3) risk of bias in classification of interventions; (4) risk of bias due to deviations from intended interventions (unexpected results); (5) risk of bias due to missing data; (6) risk of bias arising from measurement of outcomes (assessment parameters, validation protocol, independent testing protocols); (7) risk of bias in selection of reported results.

Each risk of bias is rated with "N"—No, "PN"—Probably No, "PY"—Probably Yes, and "Y"—Yes. Most studies are well-designed and have low risks in most criteria while some studies with small sample sizes have the risk of bias due to confounding, selection of participants, and measurement of outcomes. Studies with at least two "PY"s are rated "Moderate" in the summary. Ratings of individual studies are listed in Table A1.

**Table A1.** Risk of bias analysis.

| Study | Confounding | Selection of Participants | Classification of Interventions | Deviations from Intended Interventions | Missing Data | Measurement of Outcomes | Selection of Reported Results | Summary |
|-------|-------------|---------------------------|---------------------------------|-----------------------------------------|--------------|-------------------------|-------------------------------|---------|
| [79] | PN | PY | N | N | N | PY | N | Moderate |
| [80] | N | PY | N | N | N | PY | N | Moderate |
| [51] | PN | N | N | N | N | PY | N | Low |
| [81] | PN | PY | N | N | N | PY | N | Moderate |
| [53] | PN | PN | N | N | N | PY | N | Low |
| [52] | PN | PY | N | N | N | PY | N | Moderate |
| [55] | PN | N | N | N | N | PY | N | Low |
| [82] | PN | N | N | N | N | PY | N | Low |
| [69] | PN | PY | N | N | N | PY | N | Moderate |
| [74] | N | PY | N | N | N | PY | N | Moderate |
| [83] | PN | N | N | N | N | PY | N | Low |
| [84] | PN | N | N | N | N | PY | N | Low |
| [85] | N | PY | N | N | N | PY | N | Moderate |
| [76] | PN | N | N | N | N | PY | N | Low |
| [54] | PN | N | N | N | N | N | N | Low |
| [73] | PN | N | N | N | N | PY | N | Low |
| [86] | N | PN | N | N | N | PY | N | Low |
| [75] | PN | PN | N | N | N | PY | N | Low |
| [87] | PN | N | N | PY | N | PY | N | Moderate |
| [88] | PN | PN | N | N | N | PY | N | Low |
| [78] | PN | N | N | N | N | N | N | Low |
| [89] | PN | N | N | N | N | PY | N | Low |
| [90] | PN | N | N | N | N | PY | N | Low |
| [91] | PN | PY | N | N | N | N | N | Low |
| [77] | PN | N | N | N | N | PY | N | Low |
| [92] | N | PY | N | N | N | PY | N | Moderate |
| [93] | N | PN | N | N | N | PY | N | Low |
| [57] | N | PY | N | N | N | PY | N | Moderate |
| [61] | N | PY | N | N | N | PY | N | Moderate |

**Table A1.** *Cont.*

| Study | Confounding | Selection of Participants | Classification of Interventions | Deviations from Intended Interventions | Missing Data | Measurement of Outcomes | Selection of Reported Results | Summary |
|---|---|---|---|---|---|---|---|---|
| [62] | PN | N | N | N | N | PY | N | Low |
| [58] | N | PY | N | N | N | PY | N | Moderate |
| [70] | N | PN | N | N | N | PY | N | Low |
| [59] | N | PY | N | N | N | PY | N | Moderate |
| [60] | N | PY | N | N | N | PY | N | Moderate |
| [94] | N | PY | N | N | N | PY | N | Moderate |
| [63] | N | PY | N | N | N | PY | N | Moderate |
| [64] | N | PN | N | N | N | PY | N | Low |
| [65] | N | PN | N | N | N | PY | N | Low |
| [71] | N | PN | N | N | N | PY | N | Low |
| [95] | PY | PY | N | N | N | PY | N | Moderate |
| [72] | N | N | N | N | N | PY | N | Low |
| [66] | N | PN | N | N | N | PY | N | Low |
| [96] | N | PY | N | N | N | PY | N | Moderate |
| [67] | N | PY | N | N | N | PY | N | Moderate |
| [68] | N | PY | N | N | N | PY | N | Moderate |
| [104] | N | PN | N | N | N | PY | N | Low |
| [105] | N | PY | N | N | N | PY | N | Moderate |
| [106] | N | PY | N | N | N | PY | N | Moderate |
| [99] | N | PY | N | N | N | PY | N | Moderate |
| [107] | N | PN | N | N | N | PY | N | Low |
| [98] | N | PY | N | N | N | PY | N | Moderate |
| [108] | N | PY | N | N | N | PY | N | Moderate |
| [101] | N | PN | N | N | N | PY | N | Low |
| [109] | N | PY | N | N | N | PY | N | Moderate |
| [110] | N | PY | N | N | N | PY | N | Moderate |
| [111] | N | PY | N | N | N | PY | N | Moderate |
| [112] | N | PY | N | N | N | PN | N | Low |
| [113] | N | PY | N | N | N | PY | N | Moderate |
| [114] | N | PY | N | N | N | PY | N | Moderate |
| [25] | N | PY | N | N | N | PY | N | Moderate |
| [102] | N | PN | N | N | N | PY | N | Low |
| [115] | PN | PN | N | N | N | PY | N | Low |
| [116] | N | PY | N | N | N | PY | N | Moderate |
| [117] | N | PY | N | N | N | PN | N | Low |
| [118] | N | PN | N | N | N | PY | N | Low |
| [103] | PN | PN | N | N | N | PY | N | Low |
| [119] | N | PY | N | N | N | PN | N | Low |
| [120] | N | PY | N | N | N | PY | N | Moderate |
| [121] | N | PY | N | N | N | PY | N | Moderate |
| [122] | PN | PN | N | N | N | PY | N | Low |
| [123] | N | PN | N | N | N | PY | N | Low |
| [124] | N | PY | N | N | N | PN | N | Low |

**Table A1.** *Cont.*

| Study | Confounding | Selection of Participants | Classification of Interventions | Deviations from Intended Interventions | Missing Data | Measurement of Outcomes | Selection of Reported Results | Summary |
|-------|-------------|--------------------------|--------------------------------|----------------------------------------|--------------|-------------------------|-------------------------------|---------|
| [125] | N | PN | N | N | N | PY | N | Low |
| [126] | N | PN | N | N | N | PY | N | Low |
| [100] | N | PN | N | N | N | PY | N | Low |
| [97] | N | PN | N | N | N | PY | N | Low |
| [84] | N | PN | N | N | N | PY | N | Low |
| [130] | N | N | N | N | N | PY | N | Low |
| [131] | N | N | N | N | N | PY | N | Low |
| [132] | PN | N | N | N | N | N | N | Low |
| [127] | N | PN | N | N | N | PY | N | Low |
| [129] | N | N | N | N | N | PY | N | Low |
| [128] | N | PY | N | N | N | PY | N | Moderate |
| [133] | N | PY | N | N | N | PY | N | Moderate |
| [134] | N | PN | N | N | N | PY | N | Low |
| [135] | N | PN | N | N | N | PY | N | Low |
| [136] | N | PN | N | N | N | PY | N | Low |
| [137] | N | N | N | N | N | PY | N | Low |
| [143] | N | PN | N | N | N | PY | N | Low |
| [145] | PN | N | N | N | N | PY | N | Low |
| [141] | PN | N | N | N | N | PY | N | Low |
| [138] | PN | N | N | N | N | PY | N | Low |
| [140] | PN | N | N | N | N | PY | N | Low |
| [142] | PN | N | N | N | N | N | N | Low |
| [146] | N | PY | N | N | N | PY | N | Moderate |
| [147] | N | PY | N | N | N | PY | N | Moderate |
| [144] | N | PY | N | N | N | PY | N | Moderate |
| [139] | PN | N | N | N | N | PY | N | Low |
| [158] | N | PN | N | N | N | PY | N | Low |
| [159] | N | PY | N | N | N | PY | N | Moderate |
| [154] | N | PY | N | N | N | PY | N | Moderate |
| [155] | N | PY | N | N | N | PY | N | Moderate |
| [156] | N | N | N | N | N | PY | N | Low |
| [149] | N | PN | N | N | N | PY | N | Low |
| [157] | N | PN | N | N | N | PY | N | Low |
| [150] | PN | N | N | N | N | PY | N | Low |
| [153] | PN | N | N | N | N | PY | N | Low |
| [151] | N | PN | N | N | N | PY | N | Low |
| [152] | PN | N | N | N | N | PY | N | Low |
| [160] | N | PY | N | N | N | PY | N | Moderate |
| [24] | N | PN | N | N | N | PY | N | Low |
| [161] | N | PN | N | N | N | PN | N | Low |
| [148] | N | PN | N | N | N | PY | N | Low |

Abbreviations: N—No, PN—Probably No, PY—Probably Yes.

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
