# Peer review of "Applications of Deep Learning to Neurodevelopment in Pediatric Imaging: Achievements and Challenges"

_applsci, doi:10.3390/app13042302_

Round 1

Reviewer 1 Report

The review paper is mostly written in tabular format. The idea is not bad but really lacks clear motivation and strength. Please see below the main concerns:

- Section 2 should be the comparison with other reviews. What are the existing reviews regarding this topic? Why did the previous reviews fail, necessitating this new review? This is the first question, and it is necessary to justify the story behind this new review.

- The second objection is the section "Discussion". This section should find problems with the current research and give future directions.

- The third objection is somewhat similar to the second one. What are the research gaps in deep learning for neurodevelopment? The main idea of a review paper is to find the current research gap in the considered topic. How can someone find information to focus on what is missing and start new research?

- Section 2.1 should not be deep learning. This should be called machine learning. The division of supervised, unsupervised, and semi-supervised belongs to machine learning.

- In pg. 5, "GAN consists of one generator network which captures the data distribution in real images and generates a fake image and one discriminator which classifies the generated fake images and real images 5." -- before "5", it must be Figure 5.

- In pg 12, the preprocessing method name is "python" and "registration". Is it mistakes or typo? Please re-check all such problems.

- There are big problems with the reference section. The style is not consistent. The journal name is sometimes in uppercase and sometimes lowercase; please be consistent. Also, the author names are not consistent. Sometimes they are abbreviated (see numbers 114, 115, 116, and so on). Also check the reference number 157, where the author name or title is mixed and not clear.

Author Response

Kindly refer to the attached Word Document for point-by-point response and changes to the revised manuscript.

Reviewer 2 Report

This is a review study on Machine Learning (ML) applications in patients with neurodevelopmental disorders. It is a well-written paper with explanatory figures and text that is easy to follow for experts in the field but also for less experienced readers. The authors focus their analysis and discuss three ML approaches, the Multi-layer Perceptron (MLP), the Convolutional Neural Network (CNN), and the Generative Adversarial Network (GAN), and their applications in the field of neurodevelopmental medicine. They use the term "Neurodevelopment," which is an "umbrella term" containing different disorders that this work touches on. It is a challenging work with some methodological concerns that adds to the current literature touching on a rapidly developing topic. I recommend considering this study for publication after major revision addressing those concerns described below.

The term Neurodevelopmental disorder is an umbrella term that usually includes various specific diseases. 

 The main ones are Autism and ADHD, while primary structural brain abnormalities might also be represented in that category, although this might be less common. For example, it is likely that papers describing tuberous sclerosis or posterior fossa tumors may not have used the term Neurodevelopmental disorders in the mesh terms/keywords or title, which would lead to their underrepresentation in this report. The authors report that in their search, they encountered studies on the following pathologies: cerebellar dysplasia, dyslexia, epilepsy, conduct disorder, disruptive behavior disorder, post-traumatic stress disorder, tuberous sclerosis, and brain tumors.

Those disorders are very different entities, with significantly different mechanisms leading to them, which often also reflect the MRI findings of those patients. 

Comment 1:

I recommend that the authors select the disorders that they will analyze and include those in their research. Those disorders should include more than Autism and ADHD since recent reviews on those already exist in the literature.

Some examples include but are not limited to cerebral palsy, tuberous sclerosis, and disorders such as those that the authors encountered in their search. 

(The authors can consider avoiding including epilepsy as a major term to look for since this disease also entails many syndromes and pathologies that would significantly increase the volume and the complexity of this paper)

Comment 2:

The authors can consider breaking down each section to a disorder-specific approach or at least comment on each disorder separately when presenting and ideally when also discussing the results.

Comment 3:

The study did not include two big databases (ScienceDirect and Google Scholar) because of the abundance of results found after their search. This is an understandable but major limitation of the study. 

In the case that the authors decide to finally not to include those databases, they should mention that in the Limitations section of the study. Please include the limitations of the study in the paper and consider discussing the future directions of this work in the same section.

Author Response

(The authors gave the same response as above.)

Round 2

Reviewer 1 Report

Comments are taken care of. Great works.

Author Response

Thank you very much for accepting all our responses and revisions. 

Reviewer 2 Report

The authors adequately addressed the comments of the first revision of the study and the structured of the presented data has improved. This work attempts to summarize a rapidly evolving field and its value for the scientific community is high. This study can be considered for publication after addressing the following major and minor comments:

Major comments:

Comment 1:Please describe the methods that was used to identify the risk of bias in individual studies included in this review.

A method that the authors could use is the Grading of Recommendations, Assessment, Development, and Evaluation (GRADE) methodology system.

“Schünemann H, Brożek J, Guyatt G, et al; The GRADE Working Group: GRADE Handbook for Grading Quality of Evidence and Strength of Recommendations, 2013. Available at: guidelinedevelopment.org/handbook. Accessed June 1, 2020”

-Please add an additional column on the tables or a separate table with the bias grading for each study.

-Consider including the different biases that you find in the studies in the limitations section of the manuscript.

The authors have partially commented on some limitations in lines 220-225 although given the systematic review design of the study it is important that a systematic approach is used for assessing the different biases of the included studies.

Comment 2:

Please have 2 independent researchers perform this analysis to increase the internal validity for the study. If this is not possible, please include a relevant statement of the number of researchers that independently assessed the data and performed the analysis. If more than one researcher performed the research how were conflicts between their findings addressed?

Comment 3:

We suggest the researcher to register their review protocol to an international database with a suggestion being the International Prospective Register of Systematic Reviews (www.crd.york.ac.uk/prospero/)

Comment 4:

The number of studies included in tables 2 and 3 are 100 while on Figure 6 it is stated that 104 studies are reported in the review. I would kindly ask the authors to double check the studies including at the tables first for possible duplicates and secondly to add any missing studies.

Minor comments:

Comment 1:

Following the PRISMA criteria, did the authors contact with study authors to identify additional studies. If yes, please include a statement, If not please explain.

Comment 2:

Please describe the method of data extraction from the articles that you selected. What is the information and different variables that you were looking to find in each article.

-Describe that in the method’s section except for just presenting it on the tables.

-Please describe any secondary information you were interested in, and you extracted from the articles that might not be reported in Tables 2 and 3.

Comment 3:

Line 192. A better word to use to describe autism is “behaviors” instead of “activities”. Consider changing this sentence to “… restrictive and repetitive behaviors”

Comment 4:

Line 204. Grammar error: Please change to “This review paper focuses on…)

Comment 5:

Lines 207-213 Please add references.

Author Response

Kindly refer to the attached file for a point-by-point response to all comments.
